# The mechanism of eutectic growth in highly anisotropic materials

Ashwin J. Shahani[1], Xianghui Xiao[2] & Peter W. Voorhees[1]

In the past 50 years, there has been increasing interest—both theoretically and experimentally—in the problem of pattern formation of a moving boundary, such as a solidification front. One example of pattern formation is that of irregular eutectic solidification, in which the solid–liquid interface is non-isothermal and the interphase spacing varies in ways that are poorly understood. Here, we identify the growth mode of irregular eutectics, using reconstructions from four-dimensional (that is, time and space resolved) X-ray microtomography. Our results show that the eutectic growth process can be markedly different from that seen in previously used model systems and theories based on the *ex situ* analysis of microstructure. In light of our experimental findings, we present a coherent growth model of irregular eutectic solidification.

[1] Department of Materials Science and Engineering, Northwestern University, Evanston, Illinois 60208, USA. [2] X-ray Science Division, Advanced Photon Source, Argonne National Laboratory, Lemont, Illinois 60439, USA. Correspondence and requests for materials should be addressed to A.J.S. (email: shahani@u.northwestern.edu).

Far from equilibrium, solids may crystallize into highly ordered patterns, or eutectics, of remarkable complexity. Eutectic systems are ubiquitous in nature, and have been discovered in a vast array of organic[1–3], metallic[4,5] and semi-metallic alloys[6–13]. Such materials can exhibit outstanding mechanical and electrical properties because their micro-structures act as natural or *in situ* composite materials[14,15]. To tune the eutectic microstructures to technological demands, we must understand the fundamental processes underlying their formation from a featureless liquid. An exploration of this crystallization process has the potential to provide the models necessary to accelerate the design of new advanced alloys, thereby fulfilling the promise of the Materials Genome Initiative[16].

The eutectic morphologies that may arise during solidification can be classified as either regular or irregular. Regularity refers to the periodic arrangement of lamellae, and is typical of non-faceted systems. The situation is more complex when irregular eutectics are considered, in which one of the phases is faceted (for example, Si and Ge) and the other non-faceted (for example, Al and Ag). The faceted phase does not easily change direction due to its atomic structure, covalent bonding and defect-mediated growth mechanism[2,17]. Due to this inherent 'stiffness' of the faceted phase, the microstructure is non-periodic with varying interphase spacing.

In 1980, Fisher and Kurz[1] provided the first model of irregular eutectic growth. In particular, they considered the growth process of succinonitrile-borneol and camphor-napthalene irregular eutectic systems; from their experimental observations, they were able to deduce that (1) the duplex solid–liquid front is markedly non-isothermal; (2) the faceted phase leads the solidification event, that is, the faceted phase has lower undercooling and extends deeper into the melt; and (3) the non-faceted phase lags behind but spreads up the sides of the faceted phase to near its growth edges, see Fig. 1a. This model has been disputed by Hogan and coworkers[10], who suggest instead that the non-faceted phase may nucleate heterogeneously and repeatedly on exposed surfaces of the faceted flakes. According to these authors, the 're-nucleation' of grains accounts for the observed polycrystallinity of the non-faceted phase in irregular eutectic alloys[11–13].

While great insights have been gained into the structure and crystallography of irregular eutectics, their interfacial dynamics have remained an enigma for the past 50 years. Previous studies have been limited to 'quench-and-look' experiments wherein a completely solid alloy sample is analysed *post mortem*, such as the eutectic Al-Si alloy sample shown in Fig. 1b (ref. 6). However, it is well known that the quenching needed to convert the liquid to solid can distort the morphology of the solid–liquid interface from that present during crystallization. Thus, it is only through *in situ* experiments that interfacial dynamics can be tracked with a high degree of precision. To circumvent these challenges, and as mentioned above, several researchers have investigated via optical microscopy the growth of transparent organic films sandwiched between two glass slides[1–3]. Yet many details of the microstructural evolution remain unclear because of the effects of the constraint imposed by the thin film on an inherently 3D phenomenon. Furthermore, the ability of organic materials to accurately mimic the growth process of faceted phases has not been ascertained.

For these reasons, we employ a fully 4D analysis in order to investigate the dynamics of irregular eutectic growth. X-ray microtomography (XRT) is a nondestructive method to determine the evolution of a microstructure in 3D and as a function of time. In this work, we have succeeded in tracking *in situ* the growth dynamics of an irregular eutectic alloy via synchrotron-based XRT. The predominant challenge with the tomographic imaging of eutectic solidification is associated with the length scale of the lamellae ($\sim 1$–3 µm), as well as the rate at which these lamellae evolve in time ($\sim 1$–10 µm s$^{-1}$). To this end, we meet the necessary spatial and temporal resolution requirements by using the time-interlaced model-based iterative reconstruction methodology, described by Mohan and coworkers[18,19]. On the basis of our time-resolved experimental data, we determine that none of the existing models[1,7,10] are fully adequate for describing the growth mode of irregular eutectics. We find that crystallographic and topological defects in the faceted phase lead to a 'decoupling' of the eutectic constituents at the growth front.

## Results

**Tomographic reconstructions.** We investigate solidification in Al-Ge alloys of eutectic composition (51.6 wt%Ge). The Al-Ge system is an ideal candidate for XRT studies for two reasons, (1) the Ge constituent has a similar entropy of fusion as several faceted materials (for example, Si, Sb and Ga)[20], which means that we can generalize the results of our study to a host of other irregular eutectic alloys; and (2) the relatively high content of Ge provides good interphase contrast due to the large difference in atomic number between Al and Ge. The absorption-contrast XRT

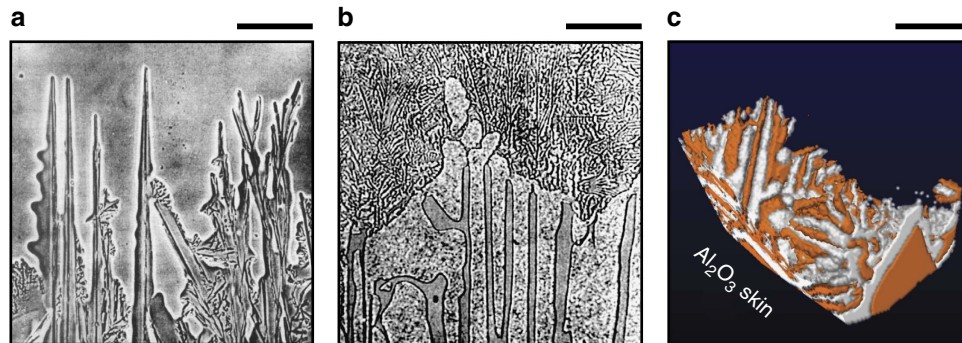

**Figure 1 | Comparison of irregular eutectic growth morphologies.** (**a**) Growth front of a eutectic mixture of borneol (faceted minor constituent) and succinonitrile[20]. Reproduced with permission from Wiley Verlag[20]. (**b**) Quenched growth front of an Al-12.7 wt%Si eutectic, where an envelope of Al encloses the faceted Si crystals[6]. According to Hellawell, this metallic 'skin' could not have been present at temperature and must have formed during the quench[6]. Reproduced with permission from Elsevier[6]. (**c**) Growth front of an Al-51.6 wt%Ge, this work. The eutectic grows from the Al$_2$O$_3$ oxide skin (not pictured) into the melt. The solid–liquid interfaces in **c** share some semblance to that of (**a**) in that the faceted Ge phase (orange) leads at the growth front. Scale bar, 12 µm (**a**), 35 µm (**b**) and 40 µm (**c**).

experiments were conducted at sector 2-BM at the Advanced Photon Source at Argonne National Laboratory. In the experiment, we cool a 1 mm diameter rod sample from above the eutectic temperature (420 °C) to 3 °C below. The sample is held at this temperature isothermally while X-ray projections are recorded continuously. The minimal 'driving force' or undercooling of 3 °C ensures that we can temporally and spatially resolve the interfacial dynamics.

Figure 2a gives a snapshot of the full region-of-interest at 180 s after the start of solidification. Shown are Ge in orange, Al in white, the melt in dark blue and the crucible walls in translucent-gray. One eutectic colony is boxed and isolated for subsequent analysis. By definition, a 'colony' refers to the portion of the microstructure that nucleated at a single site[21]. The front (0°), back (180°) and side (90°) snapshots of this eutectic colony are displayed in Fig. 2b–d. The outline of the colony is rectangular, indicating that the anisotropy of the faceted phase is an important factor during the growth process. Interestingly, the faceted Ge lamellae may be arranged in a regular pattern, see the arrow in Fig. 2d, anomalous for an irregular eutectic alloy. In general, the side profiles (Figs 1c and 2d) resemble that of the Kurz–Fisher model[1] (Fig. 1a) in that a volume of liquid is trapped between highly branched plates of Ge that extend deeper into the melt. On the other hand, the front and back views in Fig. 2b,c appear to support the 're-nucleation' hypothesis[10], wherein pockets of Al cover the exposed Ge surfaces. We reconcile these two competing viewpoints by tracking the growth of the eutectic colony in 4D.

Figure 3 shows the morphology of the eutectic colony at five representative time-steps during the growth process. As each time-step, we view the colony from its front (0°), back (180°) and side (90°), as before. While plates of Ge are evident at the earlier stages of growth, bulbous-like domains of Al envelope its surfaces at longer times. One might suppose that these bulbs of Al impede the growth of the colony in the lateral (horizontal) direction, since the Ge crystals cannot grow through the solid metal layer. However, even before the emergence of the outermost bulbs of Al, the colony extends very slowly in the lateral direction. In fact, throughout the growth process, interfacial velocities are roughly 10 times lower in the lateral direction compared with the transverse (vertical), see Fig. 4. Any nonzero growth rate in the lateral direction at long times may be due to the growth and coarsening of the outermost Al bulbs, since the Ge plates do not become appreciably thicker.

**Topological defects**. The above trends can be rationalized by considering the critical role of defects during irregular eutectic solidification. Both Si and Ge have relatively low stacking fault energies[22,23], and are thus expected to twin frequently. For instance, when viewed from the front, plates of Ge that run parallel to the transverse axis are filled with holes or gaps; these defects may result from errors in the stacking sequence of {111} planes. As shown in Fig. 2e, the kinetically mobile metallic phase can then grow through these holes and spread across the facets.

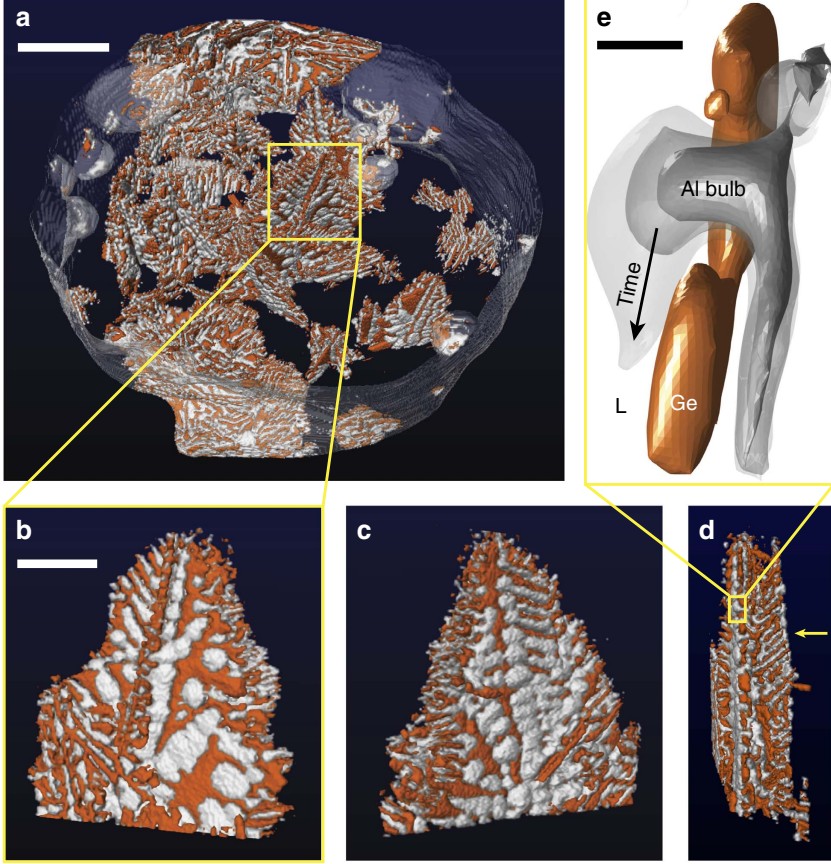

**Figure 2 | Irregular eutectic microstructures across length scales.** (**a**) Snapshot of the full tomographic region-of-interest during the growth process, where Ge is orange, Al is white and the melt is blue. One eutectic colony is boxed and isolated for subsequent 3D imaging. (**b–d**) The front (0°), back (180°), and side (90°) views of this eutectic colony. The arrow in **d** points to a region of high structural regularity, wherein the Ge lamellae are nearly parallel. The front and back views (**b,c**) suggest that pockets of Al cover the exposed Ge surfaces. (**e**) In some cases, defects cause holes or gaps within the Ge plates; then, the Al bulbs can grow through these holes and spread across Ge {111}. Scale bar, 200 μm (**a**), 70 μm (**b**) and 10 μm (**e**).

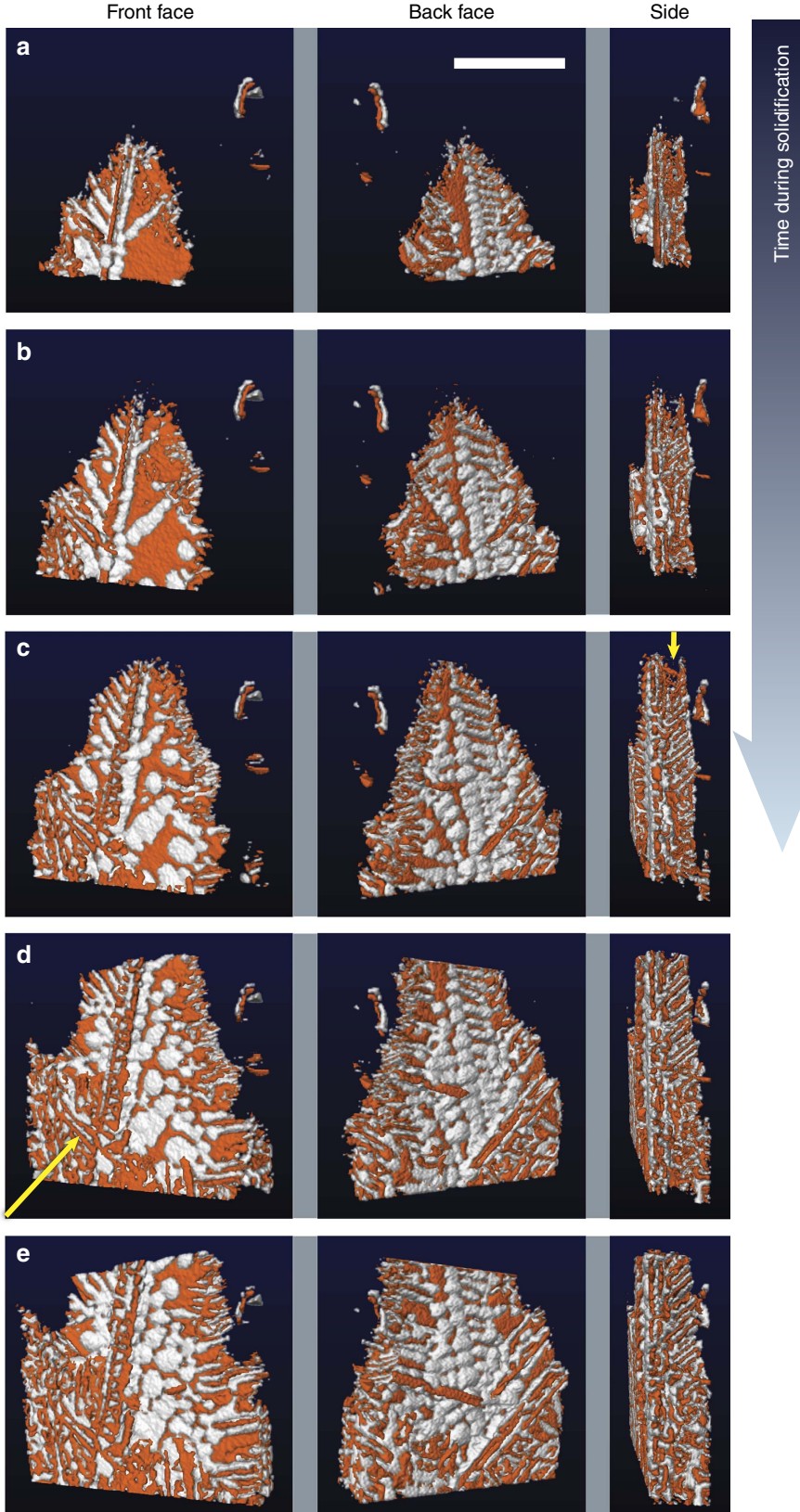

**Figure 3 | Morphology of the eutectic colony during the growth process.** Reconstructions given at (**a**) 100, (**b**) 140, (**c**) 180, (**d**) 220 and (**e**) 260 s after the start of solidification. Shown are three views per time-step, corresponding to the front (0°), back (180°) and side (90°) of the eutectic colony. The arrow in the side view of **c** points to the tips of the Ge plates that lead the solidification event; the arrow in the front view of **d** points to another eutectic colony that impinges upon the one of interest. When viewed from either the front or the back, the interfacial morphology is markedly different from that predicted by Fisher and Kurz[1]: bulbous-like domains of Al envelope the surfaces of Ge as solidification proceeds. Scale bar, 100 μm.

An important distinction between the 're-nucleation' hypo-thesis[10] and our results is that the metallic phase is not nucleating on the plates; rather, the metal emerges from holes within the plates. The contact angle at the trijunction between the Al-Ge-liquid is $> 0°$, and thus the Al spreads across the Ge plates as isolated domains. This defect-mediated growth mechanism is responsible for the bulbs of Al that cover the exposed Ge surfaces, when the colony is viewed from the front (Figs 2b and 3).

In order to focus exclusively on the Ge phase, we strip away the Al from the tomography images (Fig. 5a). Five representative holes in the frontal plate of Ge have been outlined in green for clarity. The heavily twinned microstructure of Ge in this study resembles closely that of an extracted Si flake from a deep-etched specimen[6] (Fig. 5b). This suggests that the holes are not caused by the etching procedure but are rather intrinsic to the defect structure of the faceted constituent. In addition, the holes in the

Ge phase are reminiscent of those seen in the faceted $Al_5FeSi$ phase in $Al-Al_5FeSi$ irregular eutectic alloys[24–26] (Fig. 5c). Similar to Si and Ge, the $Al_5FeSi$ phase is highly defective, with both twin and antiphase boundaries[27,28]. However, the holes and complex branching patterns observed in Fig. 5c result from the physical interaction with obstacles, for example, Al dendrites, and not necessarily from growth accidents. Thus, the solidification process of the $Al-Al_5FeSi$ eutectic is divorced, due to the separate formation of the two eutectic phases[25]. In contrast, the Al and Ge phases in our work originate within the same eutectic colony, and the branching events of Ge are crystallographically related, as will be demonstrated below. Nevertheless, both examples show the importance of crystallographic imperfections and topological defects in the development of microstructure in these materials.

**Crystallographic defects**. Defects also lead to the formation of complex regular crystals, in which parallel lamellae connect to a common spine. Such geometrical structures have been docu-mented in Si, Ge and Bi[6,8,9,17,29,30]. According to Hellawell's 'double twin' model, a {100} spine is related to {111} lamellae by twinning across axial and non-axial {111} planes, thereby producing an angle of 62.5° between lamellae and spine[6,8,9]. Evidence for twinning comes from optical metallography, in which cusps are observed where the lamellae join the interconnecting spine[6,8,9]. The periodicity of the lamellae is set by the spacing of the twins along the {100} spine. In this work, we measure the angle between lamellae and transverse spine as $61.4 ± 1.7°$ (see Supplementary Fig. 1), which agrees well with Hellawell's model. From this angular relationship, we index the spine as {100} and lamellae as {111}. Since individual lamellae are planar and run the length of the spine, the {100} spine itself is likely defect-free. As the tip of the Ge {100} spine grows unhindered in the transverse (vertical) direction, it simultaneously undergoes branching events, see the arrow in the side view presented in Fig. 3c. Initially, the Al is trailing behind the Ge {100} spine and its branches. Worth noting is that the lamellar branches are untwinned within the plate, as confirmed in back-reflection Laue photographs[8]. Because there is no such growth advantage for the lamellae, the metal eventually overgrows and engulfs the faceted phase. For this reason, we see bulbs of Al cap the lamellae on the back side of the colony (Figs 2c and 3). The sluggish kinetics of the faceted lamellae at early times, together with the Al bulbs impeding growth at later times, explain why the colony extends very slowly in the lateral direction throughout the growth process (Fig. 4). This second

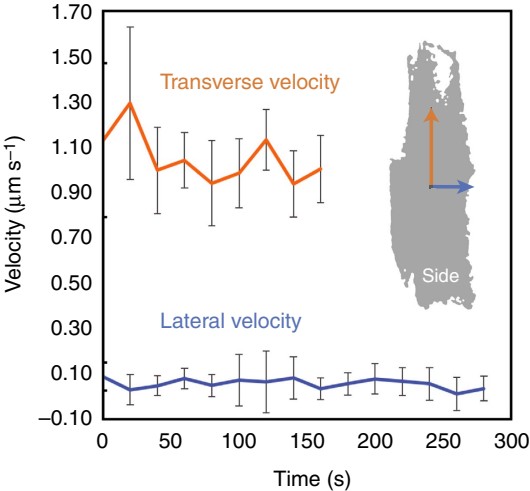

**Figure 4 | Interfacial velocities of the eutectic colony during solidification.** Measured are the growth rates in the transverse and lateral directions, as indicated by the arrows on the side (90°) silhouette of the colony (see inset). Throughout the growth process, the colony evolves roughly ten times more rapidly in the transverse direction compared to the lateral. The colony extends very slowly in the lateral direction due to the sluggish kinetics of the untwinned {111} lamellae at early times and the bulbs of Al impeding growth at later times. In the transverse direction, the Ge flakes precede the metal and are able to grow unhindered. Error bars represent one s.d., with five measurements.

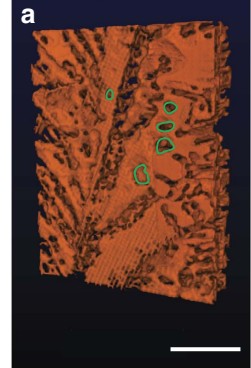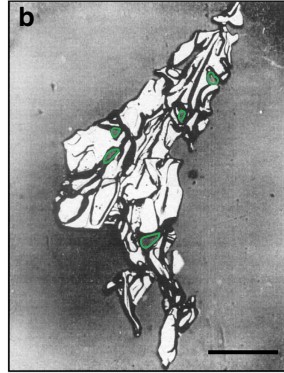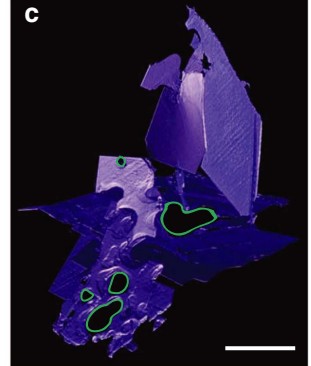

**Figure 5 | Microstructure of the faceted phase in irregular eutectic alloys.** In each of the three cases given, five representative gaps or holes are circled in green for clarity. (**a**) Front view of the eutectic colony showing the Ge phase in orange, this work. (**b**) Eutectic Si flake, extracted from a deep-etched Al-Si specimen by Hellawell[6]. The black lines correspond to multiple twin traces. Reproduced with permission from Elsevier[6]. (**c**) Eutectic $Al_5FeSi$ plates in an Al-Fe-Si alloy, the crystallization of which was imaged *via in situ* XRT by Terzi *et al.*[25] Reproduced with permission from Elsevier[25]. Scale bar 50 μm (**a**), 200 μm (**b**) and 150 μm (**c**).

mechanism of defect-mediated growth is summarized in Supplementary Fig. 2.

The above insights suggest that twinning leads to holes within Ge plates, as well as branching events between Ge plates. In both cases, defects bring about the 'decoupling' of the Al and Ge phases at the growth front, such that steady-state growth can no longer be maintained. When this occurs, the metallic phase precedes the faceted phase. Only when two or more twin planes are parallel to the growth direction, as in the twin-plane re-entrant edge mechanism[31–33], are the interfaces sufficiently mobile that cooperative growth is possible and the Fisher–Kurz model[1] is valid. Using this logic, we expect that the Ge plates grow rapidly in the transverse direction compared to the lateral (Fig. 4) because the tips of the transverse plates contain several twins that give rise to the twin-plane re-entrant edge mechanism.

In conclusion, we have investigated the growth behaviour of an irregular eutectic alloy via 4D XRT. Through these experiments we have resolved a longstanding controversy in the field, and demonstrate that none of the existing models[1,7,10] are fully adequate for describing the rich variety of anisotropic patterns that arise during crystallization. We find that defects play a critical role in the growth of the eutectic, in which pockets of metal engulf the exposed facets and control the overall growth rate of a eutectic colony. These experimental results identify a new mechanism for the growth of an irregular eutectic, and provide the key insights needed to model the crystallization of these technologically complex materials.

## Methods

**Sample preparation.** Alloy buttons of composition 48.4Al-51.6Ge (wt%) were prepared via vacuum arc-remelting at the Materials Preparation Center at Ames Laboratory, using 99.999% purity Al and 99.99% purity Ge. Then, the buttons were machined into rods measuring 1 mm diameter by 5 mm in length, using electrical discharge machining at Northwestern University. To remove any impurities that may be adsorbed onto the surfaces of the sample, the rod surfaces were etched with a 1:1 solution (by volume) of 70% $HNO_3$ and de-ionized water for 5 min.

**Beamline setup.** XRT experiments were conducted at sector 2-BM at the Advanced Photon Source at Argonne National Laboratory. The polychromatic 'pink' X-ray beam was focused on the samples and a 20 μm thick LuAg:Ce scintillator converted the transmitted X-rays to visible light. High-resolution imaging was accomplished utilizing a PCO Edge CMOS camera equipped with a $10\times$ magnifying objective to provide pixel sizes of 0.65 μm × 0.65 μm. The tomographic field-of-view measured 1,664 μm in width by 390 μm in height. The camera frame rate and exposure time were 34 Hz and 27 ms, respectively. Due to the small penetration depth through the 'heavy' element Ge (51.6 wt%), relatively long exposure times were required to ensure high signal-to-noise images.

**Experimental details.** During the experiment, the samples were heated in a resistive furnace to above the eutectic temperature (420 °C) and allowed to equilibrate. The molten specimens were held by their own oxide skin. Then, the samples were cooled to 417 °C. The specimens were held isothermally at this temperature while X-ray projections were recorded. Given the 1 mm diameter of each sample, the temperature distribution was assumed to be uniform within the sample. During acquisition, the sample was rotated continuously at a rate of 9° per second. Since the sample was rotating during the acquisition time of each projection, there may be some blurring artefact due to the sample rotation; however, this angular blur is negligibly small (0.24°).

The interlaced view sampling technique[18,19] had four sub-frames and 2,700 projections per frame. The large number of projections recorded (in addition to the high exposure time) guaranteed high-quality images. The combination of these parameters optimally allowed for a temporal discretization of 20 s between consecutive reconstructions. Data were collected for roughly 300 s, resulting in nearly four frames, 10,200 total projections and 15 total reconstructions.

**Data visualization.** Data were segmented in Matlab (MathWorks, release R2015B) using median filtering and Otsu thresholding. Examples of the reconstructed and segmented data are given in Supplementary Figs 3–4. Then, the quantized volumes were visualized in Avizo 8 (VSG).

**Data availability.** XRT projection data are stored on the Materials Data Facility (MDF) and are publicly available at http://dx.doi.org/doi:10.18126/M26P4Hhttp://dx.doi.org/doi:10.18126/M26P4H (ref. 34).

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

## Acknowledgements

This work was supported by the Multidisciplinary University Research Initiative (MURI) under award AFOSR FA9550-12-1-0458. Additional support was provided for A.J.S. by NSF Graduate Research Fellowship under grant no. DGE-1324585. The sample preparation and data acquisition were supported by the DOE under contract no. DE-FG02-99ER45782. We thank J. Sundwall and T. Bui from the Northwestern University instrument shop for machining the Al-Ge samples and the B-N crucibles. We are also grateful for helpful discussions with K.A. Mohan, E.B. Gulsoy and S.O. Poulsen. This research utilized the Quest high-performance computing facility, which is jointly supported by the Office of the Provost, the Office for Research and Northwestern University Information Technology.

## Author contributions

The experiment was conceived of by A.J.S. The XRT investigation was performed by A.J.S. and X.X. Beamline setup was performed by X.X. Data analysis was performed by A.J.S. and P.W.V.

## Additional information

**Competing financial interests:** The authors declare no competing financial interests.

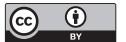

