## [Peer Review File · Nature Communications]

Reviewers' comments:

Reviewer #1 (Remarks to the Author):

The authors have proposed a new mechanism of irregular eutectic growth in Al-Ge alloy through real-time observation of the solidification using 4D X-ray microtomography imaging technique. The crystal growth morphology of non-faceted phase has been found to be different from that of faceted phase. It is therefore interesting to present the evolution of eutectic growing process with these two different phases. The authors claimed that the defects play an important role of eutectic growth, such as the holes induced by twinning of Ge plates. From what I have known in this area, the approach taken in this investigation seems to be novel, and most of the previous work in this field has been referenced.

Although the topic and figures are interesting, the present evidence falls short of providing convincing proof that irregular eutectic growth by the proposed mechanism and more details of experiments should be presented.

1) The morphology of Al phase (white) in the colony presented in Fig.2b is more like a dendritic shape rather than rectangular. The shape of the colony seems to depend more on the growth morphology of Al phase than the Ge phase (orange), thus dominated by non-faceted phase, rather than faceted phase.

2) In all three viewed faces Al phase does not totally cap the Ge plates surfaces in Fig. 3 and, hence, the slowing growth rate in the lateral direction does not rely on the obstructed growth of Ge plates by the Al envelop. The authors may mark the bulbous-like domains of Al in the figures to make their claim clearly, or presenting the top-view image of the colony for comparison as well.

3) In page 5, the author illustrate that the colony is unable to extend laterally. Through the figures presented in Fig. 3, with solidification proceeding it does extend laterally since it becomes wider. Meanwhile, the growth velocity is not zero. Here, I suspected that the phrase "unable to" is not correct to describe this growth phenomenon.

4) In Fig.3, a to e, the area fraction of the orange phase in the colony for the front face decreases with time, please explains why some Ge phase disappear. Indeed, the Ge plates should stay there until solidification completes, but it is strangely that some disappear. Is this due to the Al phase growing perpendicular to the view and throughout of the Ge phase, so that the Ge phase formed earlier was covered by the Al phase?

5) The holes or gaps formed when the Ge plates run parallel to the transverse axis, but in Fig.2 and 3 please notice these defects explicitly. The X-ray tomography is able to identify the holes formed during solidification.

6) Please provide more details and illustrations on the formation of defects and the subsequent of eutectic growth for the underlying physics that the defects formation has a direct link to the eutectic growth into irregular shape. Presently, in the paper it is not clear

as to why the eutectic growth becomes irregular due to the existence of holes. Is it just due to the decoupling growth of Al and Ge? The growth of eutectic and generation of holes would be dynamically interacted during solidification.

In spite of these weakness for the interpretations of mechanism, I still would like to recommend its publication after a major and thorough revision.

Reviewer #2 (Remarks to the Author):

The paper deals with an interesting subject, the growth of irregular eutectics. Prominent representatives are grey cast iron and AlSi alloys. Both classes of alloys have been subject to thousands of papers in the last decades and the growth of the eutectic was investigated decades ago in detail by the groups of Hellawell and Kurtz, Both teams developed different models for their microstructure evolution. These models are still used to explain main features of these materials. It is well known, that these models have their difficulties. It is really fine, that the authors re-analyse the growth of irregular euetctics with the relativley new technique of time dependent nanotomography. This technique was recently applied to AlSi, AlCu and AlSiFe alloys and gave interesting new insights into the microstructure evolution and many casting problems associated with for instance the formation of platelike AlSiFe intermetallics. Therefore the papers deals with a subject right in the heart of current scientific interest in solidification science. The approach and interpretation used by the authors is excellent and their observation shows, that an essential new feature is the appearance of defects in plates of the faceted phase while growing. They attribute such defects to errors from stacking sequence (this idea will fertilize further experimental studies to prove or find the origin of such stacking sequence erros; it reminds me of the well known problem in twinning of the faceted phase, which was recnetly solved by Schumacher an co-workers with high resolution TEM). While their hypothesis might be right, it is important, that such defects occur and then lead to growth of the non-facetted phase into these holes (such structures were observed by post-mortem tomography by Mikolaczjak and Ratke, Luc Salvo and Co-workers, Arne Dahle etc., reported in Acta Materialia and Met. Trans.; might be fine, if the authors would take a look to these papers).

In summary: the paper is very well written, deals with a fascnating subject, reveals new data for the complex pattern evolution in irregular eutectics. It definitvely will fertilize and stimulate the scientific discussion on the growth of this important class of alloys.

The paper can be published as it is.

Reviewer #3 (Remarks to the Author):

The authors describe anisotropic behavior of dendrites in Al-Ge alloy. The topic is of interest

to the metals community, but perhaps not so much to the community at large. The data presented is of high quality and the interpretations are solid.

My main concern with this paper is its appropriateness for Nature Communications. I don't believe that it is novel enough to merit publication in this journal. It would be better suited for a journal in the metals field. The techniques used are fairly standard, so the novelty does not really lie in the technique used.

More thorough quantification of the data, as well as correlation to crystallography might also be a good idea.

1 Response to Reviewer #1

1. The reviewer first comments that “the morphology of the Al phase (white) in the colony presented in Fig. 2(b) is more like a dendritic shape . . .” However, we note that this apparent dendritic shape is only coincidental; the eutectic Al grows through holes or gaps in the Ge facets. One of these gaps is long and narrow and thus the kinetically mobile Al constituent can propagate through this oblong opening. The Al may then resemble a primary dendrite arm, although the reality is quite different.
2. The reviewer’s second suggestion is to “mark the bulbous-like domains of Al in the figures.” To this end, we now label the Al bulb in Fig. 2(e). Note that there are many such bulbs, as shown in Fig. 3(e).
3. We agree that the wording “unable to extend laterally” is confusing because the colony *does* grow laterally, albeit at a very slow rate (Fig. 5). We have replaced this wording in the revision.
4. The reviewer asks why the Ge constituent “disappears” during the growth process, upon inspection of Fig. 3. However, this is not the case because the Al phase covers or envelopes the exposed Ge facets during growth, as shown in Fig. 2(e); this may give the appearance of Ge melting. We mention this growth mechanism explicitly in the text and in figure captions. This is also why a fully 4D (i.e., 3D space plus time) analysis is needed to investigate such complex growth forms.
5. The reviewer requests that we identify in our images the hole defects in the Ge plates. To this end, we have included a new figure (Fig. 4(a)) that shows the Ge constituent alone, i.e., without the Al. Here, we have outlined five representative holes in green. We hope that this addition clarifies the highly defective microstructure of the faceted constituent in irregular eutectic alloys.
6. Lastly, the reviewer comments that “it is not clear as to why the eutectic growth becomes irregular due to the existence of holes.” We note that the term *irregular* in the manuscript refers only to the fact that one of the two constituents is faceted (Ge) while the other is atomically rough (Al), see page 2. This is the hallmark of irregular eutectic alloys. Our work aims to provide a new mechanism by which irregular eutectic alloys (such as Al-Ge, Al-Si, etc.) grow from a liquid phase. As mentioned in the text, defects in the faceted constituent give rise to holes and branching events, see pages 6-7.

2 Response to Reviewer #2

We are thankful to the second reviewer for bringing to our attention the great body of work²⁶⁻²⁸ on characterizing the growth process of the Al-Al₅FeSi irregular eutectic, wherein the Al₅FeSi constituent is similarly faceted and highly defective. We have added in Fig. 4(c) to our discussion, reproduced with permission from the work by Terzi, Dahle, and coworkers.²⁷ We discuss the similarities and differences between this figure and our own (Fig. 4(a)), which represents the Ge constituent in the Al-Ge eutectic. Importantly,

“... the holes and complex branching patterns observed in Fig. 4(c) result from the physical interaction with obstacles, e.g., Al dendrites, and not necessarily from growth accidents. Thus, the solidification process of the Al-Al₅FeSi eutectic is *divorced*, due to the separate formation of the two eutectic phases.²⁷ In contrast, the Al and Ge phases in our work originate within the same eutectic colony, and the branching events of Ge are crystallographically related ...”

Thus, the Al and Ge are more strongly coupled in the Al-Ge eutectic. For this reason, Al-Ge shares some semblance to the classical picture of irregular eutectic growth; see the revised Fig. 1 where we compare past work by Fisher and Kurz,⁸ and Hellawell,¹⁰ to our own. In addition, we agree with the reviewer that this work will “stimulate the scientific discussion on the growth of this important class of alloys.”

3 Response to Reviewer #3

The third reviewer first comments that our work is “of interest to the metals community but perhaps not so much to the community at large.” As such, the reviewer believes that this work is “better suited for a journal in the metals field.” We would like to point out that eutectic patterns are in fact ubiquitous: they have been documented not only in metallic systems, but also in ceramic, polymeric, and organic mixtures, see Refs.³⁻¹⁵ within the text. In particular, the earliest *in situ* studies on irregular eutectic solidification were conducted using mixtures of succinonitrile and borneol, and camphor and naphthalene.³ This is mentioned verbatim in the manuscript (page 2). Thus, the problem of pattern formation of a moving boundary is fundamental to nearly all sub-disciplines within materials science and encompasses both hard and soft materials.

Secondly, the reviewer writes that “the techniques used are fairly standard, so the novelty does not really lie in the technique used.” To this end, we note that high resolution, 4D (i.e., space and time resolved) X-ray tomography has only recently been applied to the study of solidification phenomena, such as dendritic growth.²¹ For the first time, we can measure the dynamics of solid-liquid interfaces with sub-micrometer and sub-second resolution. Such studies would otherwise not have been possible without the great strides in data sampling and reconstruction, computer hardware and storage, and algorithms for processing Big Data in a massively parallel environment. More specifically, the time-interlaced, model-based iterative reconstruction (TIMBIR) algorithm applied in this work to achieve high quality reconstructions was published just one year ago.²⁰ Thus, high spatial and temporal resolution 4D X-ray tomography has opened a new paradigm in the field of solidification science.

REVIEWERS' COMMENTS:

Reviewer #1 (Remarks to the Author):

The authors have improved the manuscript. Most of the major concerns have been addressed. I recommend to accept the publication of the paper.